# Table Tennis Tutor: Forehand Strokes Classification Based on Multimodal Data and Neural Networks

**DOI:** 10.3390/s21093121

**Published:** 2021-04-30

**Authors:** Khaleel Asyraaf Mat Sanusi, Daniele Di Mitri, Bibeg Limbu, Roland Klemke

**Affiliations:** 1Cologne Game Lab, TH Köln, 51063 Cologne, Germany; rk@colognegamelab.de; 2DIPF|Leibniz Institute for Research and Information in Education, 60323 Frankfurt, Germany; dimitri@dipf.de; 3Leiden Delft Erasmus-Center for Education and Learning, Technical University Delft, 2628 CD Delft, The Netherlands; b.h.limbu@tudelft.nl; 4Technology-Enhanced Learning & Innovation, Open University of the Netherlands, 6419 AT Heerlen, The Netherlands

**Keywords:** multimodal data, neural networks, psychomotor learning, table tennis, activity recognition, sensors, learning analytics

## Abstract

Beginner table-tennis players require constant real-time feedback while learning the fundamental techniques. However, due to various constraints such as the mentor’s inability to be around all the time, expensive sensors and equipment for sports training, beginners are unable to get the immediate real-time feedback they need during training. Sensors have been widely used to train beginners and novices for various skills development, including psychomotor skills. Sensors enable the collection of multimodal data which can be utilised with machine learning to classify training mistakes, give feedback, and further improve the learning outcomes. In this paper, we introduce the Table Tennis Tutor (T3), a multi-sensor system consisting of a smartphone device with its built-in sensors for collecting motion data and a Microsoft Kinect for tracking body position. We focused on the forehand stroke mistake detection. We collected a dataset recording an experienced table tennis player performing 260 short forehand strokes (correct) and mimicking 250 long forehand strokes (mistake). We analysed and annotated the multimodal data for training a recurrent neural network that classifies correct and incorrect strokes. To investigate the accuracy level of the aforementioned sensors, three combinations were validated in this study: smartphone sensors only, the Kinect only, and both devices combined. The results of the study show that smartphone sensors alone perform sub-par than the Kinect, but similar with better precision together with the Kinect. To further strengthen T3’s potential for training, an expert interview session was held virtually with a table tennis coach to investigate the coach’s perception of having a real-time feedback system to assist beginners during training sessions. The outcome of the interview shows positive expectations and provided more inputs that can be beneficial for the future implementations of the T3.

## 1. Introduction

Table tennis or ping pong is a complex sport, requiring various psychomotor skills for different techniques and movements. It is widely known for its multiple shot techniques, spins and playing styles, played at a fast pace [1]. For beginners, it is essential to master the four fundamental strokes (forehand drive, backhand drive, forehand push, backhand push), before moving on to more complicated techniques [2]. The most fundamental technique is the forehand drive or forehand stroke, categorised into two types; (1) short-stroke and (2) long-stroke. A short-stroke is executed based on the rotation of the body to the left/forward, from the hips (based on the right-handed player), in which the endpoint of the bat shouldn’t be over the body. Otherwise, the stroke is long and considered to be one of the common beginner mistakes [3]. From an expert’s perspective, a long stroke is considered to be an advanced technique which requires the fundamentals, such as the short forehand stroke to be mastered first.

To learn the technique, beginners need to repeatedly practice the stroke to the extent that muscle memory is trained, which automates them. Doing so allows the player to execute the stroke without any mental effort, and thus he/she can focus on more complex aspects of performance. This requires repeated deliberate practice, a conscious form of practice, which necessitates constant feedback from the mentor [4]. A beginner needs continuous feedback on various attributes of the skills such as the body movement and strokes during repeated practice, from the mentor in order to improve. However, the mentor’s inability to be around all the time will adversely affect the beginners’ progress requiring longer duration to master the skill. Moreover, the beginners could also develop improper techniques that, once automated, are difficult to change [5]. Limbu et al. [6], in their work, have pursued to address this issue by using sensors to record expert models and use the expert model to train beginners. In line with their work, this study aims to address this issue of shortage of mentors for training in table tennis by using sensors to complement mentors in providing immediate real-time feedback to the beginners.

In a high-quality training facility, such sessions are supported by sports training technologies that use expensive sensors and equipment, which are unavailable to the general population. However, smartphone devices, which are widely available for the general population, are equipped with high-tech sensors, such as the accelerometer and gyroscope that can collect data in real time. These sensors in smartphone devices can be used to measure and monitor body motion to provide feedback during table tennis practice and, therefore, can be potentially used for training [7,8,9]. The majority of smartphone devices have built-in sensors that measure acceleration, orientation, and various environmental conditions. These sensors provide raw data with high precision and accuracy. In addition, they are also capable of monitoring three-dimensional device movement or positioning or monitoring changes in the ambient environment (Android Sensors Overview—https://developer.android.com/guide/topics/sensors/sensors_overview, last accessed on 20 March 2021). Data from smartphones have been used to predict human activities such as movement [10,11] and fall detection [12]. Another external sensor used for training psychomotor skills is Microsoft Kinect [13,14,15]. The Kinect is an external infrared sensor that is used to track skeletal points of the human body and their movements, which can also be useful in the case of table tennis. These aforementioned devices can be used simultaneously, enabling the collection and analysis of the multimodal data for providing feedback during table tennis training.

We define multimodal data as the data from various sensors that represent more than one modality with the intention to capture an experience as close to human perception. Multimodal data often provide a better understanding of an activity that it represents. Educational researchers in the field of technology-enhanced learning and artificial intelligence (AI) are increasingly using multimodal data for training machine learning (ML) models that enhance, out of many others, psychomotor learning  [16]. The spread of smartphones, wearable sensors, and depth cameras has made the collection and use of multimodal data feasible for the general population. In recent years, the multimodal interaction approach has become increasingly popular and widely adopted in the learning science and learning analytics domains. This can be observed by the rising interest in Multimodal Learning Analytics (MMLA), an integration of a sensor-based approach with learning analytics to better understand the learning behaviour of a student. Similarly, multiple sensors, and thereof multimodal data, are increasingly being used to facilitate multimodal learning environments, which we term as Multimodal Learning Experience (MLX). Furthermore, with multimodal data, AI/ML can better understand the students’ actions, such as psychomotor activities, and thus provide better feedback during MLX.

In this paper, we first explore how multimodal data and ML approaches can be applied for automatic mistake detection in table tennis. To do so, we prototyped the Table Tennis Tutor (T3), a multi-sensor, intelligent tutoring system consisting of a smartphone with its built-in sensors and a Microsoft Kinect. We, with the prototype, investigated whether the smartphone sensors by themselves are able to classify forehand table tennis strokes as accurately as the Kinect, using the Kinect data as a baseline for performance comparison. Then, we investigated how the teachers conceptualised a system for providing real-time feedback to the beginners, by conducting an expert interview with a table tennis coach. As such, we investigated the following research questions:RQ1Can we use the sensor data from the smartphone to classify forehand strokes in table tennis training, as accurately as external sensors, for example, Microsoft Kinect?RQ2How do table tennis coaches conceptualise a real-time system that provides immediate feedback to the beginners during training sessions?

The paper is structured as follows: in Section 2, we present related studies that utilise sensors for the collection of multimodal data to help improve the learning outcome and to what extent they are used in the pyschomotor domain. In Section 3, we describe the methodology of this study. In Section 4, we report the results of the the study, followed by a discussion of the results in Section 5. Finally, we explain the findings of the study in Section 6, limitations of the study in Section 7, and possible future work in Section 8.

## 2. Related Work

### 2.1. Sensors in Sports

Sensors are increasingly becoming portable and smarter, enabling efficient methods for the acquisition of performance data, which allows for effective monitoring and intervention. Such sensors have been explored to provide support in various learning domain [8,15,17,18,19]. Schneider et al. [20] analysed 82 prototypes found in literature studies based on Bloom’s taxonomy of learning domains and explored how they can be used to provide formative assessment, especially as a feedback tool. The analysis revealed that sensor-based learning applications could assist learning scenarios in several areas, including the psychomotor domain. This research recommends researchers and educators to consider sensor-based platforms as dependable learning tools to reduce the workload of human teachers or mentors and, therefore, contribute to the solution of many current educational challenges.

A single motion sensor, for example, the accelerometer, enables the recognition of human activities such as walking, running, jogging, etc. Pereira dos Santos et al. [8] explored how an accelerometer sensor can be utilised to capture motion data associated with critical aspects of learning in the context of social dancing. The results of the study accentuate the potential of sensors for recognising human activities. However, a single source of data is impractical in recognising more complex activities. Consequently, more sensors are needed and used in a synchronised manner, leading to a multimodal setup. In the domain of racket sports, which typically involves more complex techniques and dynamic movement, Sharma et al. [21] present a technique based on a combination of Inertial Measurement Unit and audio sensor data embedded in a smartwatch to detect various table tennis strokes. Anik et al. [22] used a motion-tracking device that contains both accelerometer and gyroscope attached to the badminton racket to recognise the different types of badminton strokes such as serve, smash, backhand, forehand, return, etc. The results from these three papers show that the combination of sensors contributes to a better recognition rate and improvement than using these sensors individually. In an approach more related to ours, Viyanon et al. [7] developed a smartphone application that analyses player’s forehand and backhand table tennis strokes using both accelerometer and gyroscope embedded in a smartphone device that is strapped on the player’s wrist. The result shows that smartphone sensors can recognise the aforementioned strokes. However, the authors discussed that having the smartphone device strapped on the player’s wrist for a long continuous session would introduce additional inconvenience.

### 2.2. Multimodal Data for Psychomotor Learning

In the psychomotor domain, motion sensors such as accelerometers and gyroscopes are predominantly used for the acquisition of motion data to recognise human activities. Pereira dos Santos et al. [8] emphasise the importance of combining motion sensors to obtain higher accuracy in detecting not only simple but complex activities as well. Combining these sensors enables the collection of multimodal data and, therefore, a more accurate representation of the learning process [16]. The collection of multimodal data can be done using various types of sensors such as wearable sensors, depth cameras, Internet of Things devices, computer logs, etc. Moreover, these sensors are typically synchronised with human activities so that one can analyse historical evidence of learning activities and the description of the learning process [23]. Limbu et al. [6] developed the “Calligraphy Tutor”, which uses Pen sensors in Microsoft Surface and EMG sensors in Myo (Myo sensor armband—https://www.rs-online.com/designspark/expanding-gesture-control-with-the-myo, last accessed on 25 March 2021) sensor armband to provide feedback to learners during practice. It also allows the calligraphy teacher to create an expert model, which the learners can later use to practice and receive guidance and feedback based on the expert model. Similarly, Schneider et al. [24] also designed a system to support the development of public speaking skills. The system uses the Microsoft Kinect v2 (Kinect v2—https://docs.depthkit.tv/docs/kinect-for-windows-v2, last accessed on 25 March 2021.) depth camera sensor that is placed in front of the learner to track the skeletal joints of the learner’s body, along with a microphone, while presenting and provide real-time feedback based on common public speaking mistakes such as facial expressions, body posture, voice volume, gestures, and pauses.

### 2.3. Supervised Learning for Classification

To better understand learners’ psychomotor performance, AI, more precisely machine learning approaches, have been explored. Machine learning models are used to classify activities based on collected multimodal data. For instance, Di Mitri et al. [17] investigated to what extent multimodal data and Neural Networks can be used for learning Cardiopulmonary Resuscitation skills by utilising a multi-sensor system consisting of a Kinect depth camera sensor for tracking body position and a Myo sensor armband for collecting electromyogram information. Furthermore, the authors used the Long Short-Term Memory (LSTM) as the machine learning model to classify the CPR activities.  The study results show a high classification rate of the activities when combining the sensors, emphasising the importance of the multimodal approach and Neural Network in classifying complex psychomotor activities. In another example, Lim et al. [25] presented a deep learning-based coaching assistant method to support the table tennis practice by using a combination of LSTM with a deep state-space model and probabilistic inference to classify five table tennis skills. For collecting the motion data, three Inertial Measurement Unit sensor modules were attached to the right hand and arm of the player. The experimental results show that the LSTM achieved a classification rate of 93.3% of accuracy.

In another supervised learning method example, Anik et al. [22] applied two machine learning model approaches that are the K-Nearest Neighbors and Support Vector Machines (SVM) classifiers using the motion data collected from a large set of users to recognise the different types of badminton strokes. Similarly, Blank et al. [26] presented an approach for table tennis stroke detection and stroke type classification using inertial sensors attached to table tennis rackets and collected data of eight different basic stroke types from ten amateur and professional players. Multiple classifiers were evaluated, and different features were extracted based on the detected strokes. Both results show that their best performance was achieved with the SVM approach. Liu et al. [27] applied the SVM to recognise five different table tennis strokes based on the motion data collected from nine athletes wearing Body Sensor Networks. Consequently, the results show that the SVM achieved an accuracy of 97.41%.

These aforementioned papers indicate that supervised learning such as the LSTM and SVM can classify activities in the psychomotor domain. Tabrizi et al. [28] performed a comparative study of classifying forehand table tennis strokes using three machine learning models: LSTM, Two-Dimensional Convolutional Neural Network, and SVM. Additionally, the motion data were collected from a BNO055 smart sensor mounted on the table tennis racket. The results consequently show that the LSTM model performed better than the other two models, thus reinforcing our intention to train a Neural Network to classify forehand strokes in table tennis using smartphone sensors.

## 3. Method

The study reported here evaluates the stroke detection system of the T3 to train table tennis using smartphone sensors. To achieve this, a workflow consisting of steps for collecting and processing the data recommended by [29] is followed. First, we explore whether the smartphone sensors alone could classify forehand table tennis strokes as accurately as the Kinect, using the Kinect data as a baseline for performance comparison. Then, we investigate how the teachers conceptualise a computer system that provides real-time feedback to beginners by conducting an expert interview with a table tennis coach.

### 3.1. Participants

Due to the COVID-19 lockdown regulations, the research team was limited from acquiring enough participants in this research. After multiple search attempts and time constraints, we managed to obtain only one participant for collecting the data and one coach for the expert interview.

For RQ1, we managed to acquire only a single male participant from the Open University of The Netherlands who has approximately 19 years of experience in playing table tennis casually. With his years of experience and sufficient knowledge, we used him to collect data for training the model. We asked him to execute the correct forehand strokes and mimic the incorrect forehand strokes repetitively. The study was not conducted in a professional playground but rather in a casual office play area with an Olympic-sized standard tennis table and balls. Due to the limited amount of balls and lack of self-serving machines, we had a volunteer to serve and return the incoming balls for longer rallies to allow efficient recordings.

Due to COVID, we could not conduct the interview (RQ2) physically. Therefore, we decided to do the it online. We contacted 13 table tennis clubs in Germany, the Netherlands, and Malaysia to help us acquire the coaches for our study. Out of these, only one female coach from the University Sports of the University of Saarland (Germany) agreed to take part in the study. The coach has 21 years of playing experience and roughly four years of coaching experience. The coach was briefed, gave her consent, and agreed with the video data recording. The data collected were fully anonymised.

### 3.2. Smartphone Placement Study

Before proceeding with RQ1, we assumed that the placement of the smartphone would likely result in a differing quality of data collected. Therefore, to figure out an ideal position for the smartphone to be placed during the training, we conducted this initial study. For this study, we considered the chest area (horizontally and vertically), arm, and pocket. These particular positions were chosen according to the recommendation of the participant about the key motion points during the execution of the forehand stroke. We excluded the wrist due to the size and weight of the device as it can be inconvenient and unnatural to the user to practice with the device strapped to the hand [7].

The positions mentioned above, except the pocket, need additional accessories to hold the smartphone device. Therefore, a harness and mobile strap are used for the chest and arm, respectively. These methods have been used for helping patients with rehabilitation exercises, which involve body motions [9]. On the other hand, placing the smartphone in the pocket has been widely utilised for recognising human activities [11]. Additionally, we were focused on providing a proof of concept within a casual table tennis playing scenario. Therefore, we do not consider the implications of having a smartphone in a loose-fitting dress commonly used in table tennis.

For the arm and pocket, the right side was initially selected because the participant was right-handed. Sets of data were collected with these four positions and compared visually by the research team using the Visual Inspection Tool (VIT) [30] (VIT—https://github.com/dimstudio/visual-inspection-tool, last accessed on 25 March 2021), an existing annotation toolkit, by observing the data plots from the time series to explore the similarity in terms of patterns when the participant performs the strokes. The visualised results for this study are shown in Figure 1.

Based on Figure 1, the first five peaks’ annotations represent short strokes performed by the participant. Long strokes were mimicked by the participant and annotated as the last five peaks. By analysing these visually, we observed that the data points for the horizontally-placed smartphone (see Figure 1a) have a unique plot pattern as compared to the other three. Hence, we excluded this position. However, for the other three positions, the plot patterns are relatively similar to each other. Reasonably, we chose the right pocket as we won’t need extra accessories to hold the smartphone, which could introduce additional inconvenience for the participant to perform the stroke. For the Kinect, we positioned the device in the middle of the tennis table (see Figure 2) in order to capture the participant’s upper body joints, which are vital for identifying the strokes.

### 3.3. Data Collection Setup

Figure 2 shows the setup used for collecting multimodal data in the case of table tennis training. It consists of the following devices/sensors:Sony Xperia Z3–a smartphone device consisting of an accelerometer and a gyroscope capable of capturing the participant’s motion data.–positioned inside the participant’s right pocket.Microsoft Kinect (v2)–a depth camera sensor capable of recording the participant’s three-dimensional skeleton position and capturing video of the participant.–placed in the middle of the tennis table.Both these devices/sensors were connected to the laptop on the side for handling data collection

In order to collect data from the participant in a systematic manner, we adopted the Multimodal Learning Analytics (MMLA) Pipeline [29], a workflow for the collection, storage, annotation, analysis, and exploitation of multimodal data for supporting learning. We developed the LHAPP4 (LHAPP4—https://github.com/khaleelasyraaf/Sensor_Data_Android.git, last accessed on 19 April 2021), a smartphone application that sends data wirelessly to the LearningHubAndroid. The LearningHubAndroid (LearningHubAndroid—https://github.com/khaleelasyraaf/LearningHubAndroid.git, last accessed on 19 April 2021) is a Windows application that receives the data from the clients such as the LHAPP4 via WLAN, which then forwards the data to the laptop that runs the Multimodal Learning Hub (LearningHub) [31]. The LearningHub is a custom-made sensor fusion framework designed to collect sensor data in real-time from different sensor applications and generate synchronised recordings in the form of zip files. The LearningHub is designed for short activities—a session should not be more than 10 min. It uses a client–server architecture with a master node controlling and receiving updates from multiple sensor or data provider applications. Each data provider application retrieves data updates from a single device, stores it into a list of frames, and, at the end of the recordings, sends the list of frames to the LearningHub. In this way, the LearningHub allows collecting data from various sensor streams produced at different frequencies.

We used existing components for the Kinect (Kinect Reader), which was also connected to the LearningHub for data collection. Additionally, we used the Visual Inspection Tool (VIT) [30] for data annotation and Sharpflow [17] for data analysis. These components above enabled the implementation of the MMLA Pipeline.

In the T3, the LearningHub was used together with three data provider applications, the LearningHubAndroid, the Kinect reader, and a screen recorder. The smartphone data provider consists of two applications: LHAPP4 and LearningHubAndroid. Thus, using these applications, we collected accelerometer and gyroscope data represented as *x*, *y*, and *z*-axes, each directional and rotational, for a total of six attributes. The Kinect Reader collected data from the participant’s upper body joints (i.e., head, hands, shoulders, elbows, and spine) that are visible to the Kinect camera represented as three-dimensional points in space and position features for a total of 40 attributes. We excluded the lower body joints (i.e., knees and ankles) due to the obstruction of the table. The screen recorder captured the video of the participant performing the forehand strokes through the point of view of the Kinect device itself. Figure 3 provides an overview of the setup, including the tools and sensors used. Highlighted in red are the new components introduced by this research. In blue are the future works for the T3. In the following sections, we describe how we utilised these components in this research.

### 3.4. Data Collection Procedure

After all the applications were ready, the participant was handed over the smartphone, which he places inside his right pocket. The participant stood on the side where the Kinect is positioned while the assistant played from the other side. The assistant served the participant, after which the two started practising short strokes or long strokes based on the intentions of data to be collected. The data were collected in multiple instances, as the LearningHub does not support long-duration recordings, and being focused consistently is mentally and physically tiring for the participant during practice. We recorded multiple sessions with the participant performing the short strokes and the long strokes repetitively. The participant dataset resulted in a total of 33 sessions and 510 recorded strokes.

### 3.5. Data Storing

The LearningHub uses a custom data format of Meaningful Learning Task (MLT-JSON session file) for data storing and exchange. The MLT session comes as a compressed folder including (1) one or multiple time-synchronised sensor recordings (JSON files); and (2) one video/audio of the recorded performance. The compressed folder consists of data of total attributes from smartphone and Kinect sensors which are serialised into JSON files with the following properties: an applicationId, an applicationName, and a list of frames (see Listing 1 and Listing 2). These frames have a timestamp and a sensor attribute with their corresponding values. Furthermore, an MP4 file with the recorded video of a session is added. In our case, each session produced a compressed file of roughly 15 Mb.

**Listing 1 sensors-21-03121-i001:**
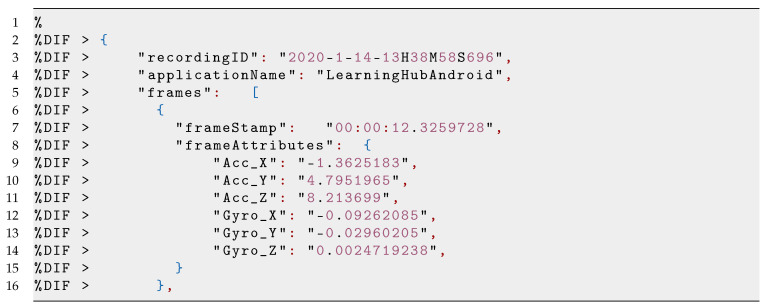
Example of the JSON string from the LearningHubAndroid containing a RecordingObject frame for each of the DataProvider Applications.

**Listing 2 sensors-21-03121-i002:**
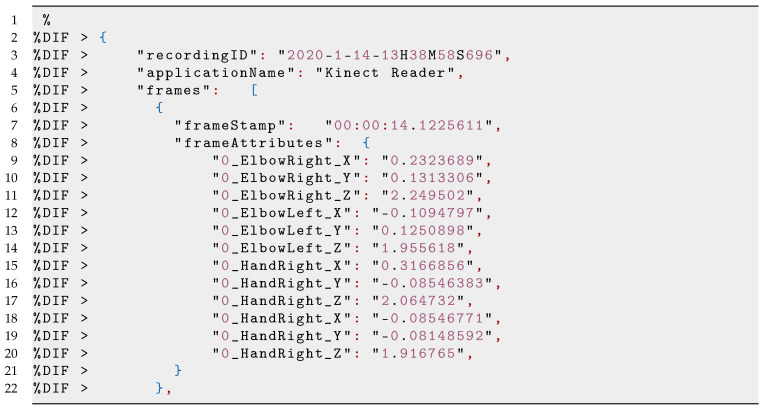
Example of the JSON string from the Kinect Reader containing a RecordingObject frame for each of the DataProvider Applications.

### 3.6. Data Annotation

The annotations were synchronised with the recorded sessions using the VIT [30], a web application prototype that allows manual and semi-automatic data annotation of MLT-JSON session files generated by the LearningHub. The VIT allows for specifying custom time-based intervals and assigned properties in those intervals.

Figure 4 shows the screenshot of the VIT used for the detection of strokes in a single session. For each of the 33 sessions recorded, the annotation file was loaded and synchronised manually with the guidance of the sensor data plots and the video recordings. We added the time-intervals to portions of sensor recordings. We used the binary classification approach, where either the stroke is correctly executed (Class 1) or incorrectly executed (Class 0) as labels for the machine learning classification (see Listing 3. Thus, a total number of 510 strokes (260 short strokes + 250 long strokes) were annotated. Subsequently, we downloaded the annotated sessions with the exclusion of the video files, which are vital for the next step, the analysis of data for the classification of strokes.

**Listing 3 sensors-21-03121-i003:**
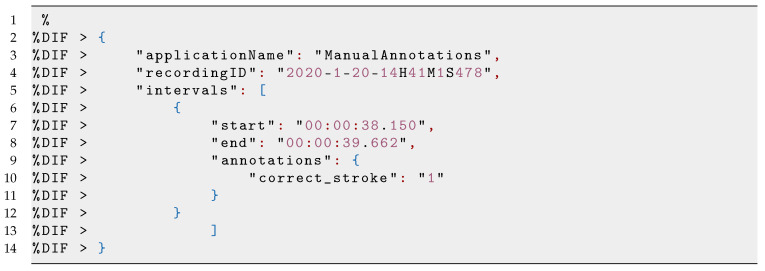
Example of the annotation file.

### 3.7. Data Analysis

We used Sharpflow (SharpFlow—https://github.com/dimstudio/SharpFlow/, last accessed on 19 April 2021), a data analysis toolkit for time-series classification using deep and Recurrent Neural Networks (RNNs). In this study, the model used for the classification is the LSTM  [32], a special kind of RNN capable of learning over long sequences of data. In the training phase, the entire MLT-JSON sessions with the inclusion of their annotations are loaded into the memory and transformed into two Pandas DataFrames (Pandas DataFrame—https://pandas.pydata.org/pandas-docs/stable/reference/api/pandas.DataFrame.html, last accessed on 20 March 2021) containing: (1) the sensor data, and (2) the annotations. Since the sensor data from two devices came with different sampling frequencies, the sensor data frame had a great number of missing values. To minimise this issue, each stroke was resampled into a fixed number of time-bins that correspond to the median length of each sample. Hence, a tensor of shape (# samples ×# bins × # attributes) was obtained. Using random shuffling, the dataset was split into 85% for training and 15% for testing. A part of the training set (15%) was used as a validation set. Additionally, we applied feature scaling using min-max normalisation with a range of –1 and 1. The scaling was fitted on the training set and applied on both validation and test sets. The implementation of the LSTM model was made using PyTorch, an open-source Torch library designed to enable rapid research on machine learning models. The architecture of the model (see Figure 5) was based on Di Mitri et al. [17], a sequence of two stacked LSTM layers followed by two dense layers:

–a first LSTM with input shape 51 × 46 (#intervals × #attributes) and 128 hidden units;–a second LSTM with 64 hidden units;–a fully connected layer with 32 units with a sigmoid activation function;–a fully connected layer with a single target class (*correct_stroke*) with a sigmoid activation function;

The original code from Di Mitri et al. [17] remains unchanged, but the configuration was fitted to our dataset (see Listing 4). Since the output of the model consists of a binary class, we use a binary cross-entropy loss for optimisation and training for 30 epochs using an Adam optimiser with a learning rate of 0.01. The number of 30 epochs was chosen after acknowledging the overfitting turning point, i.e., the point when validation loss values stop increasing.

The LSTM network was trained and validated three times: (1) considering just the smartphone data as the input of the model, (2) just the Kinect data, and (3) smartphone and Kinect data combined.

**Listing 4 sensors-21-03121-i004:**
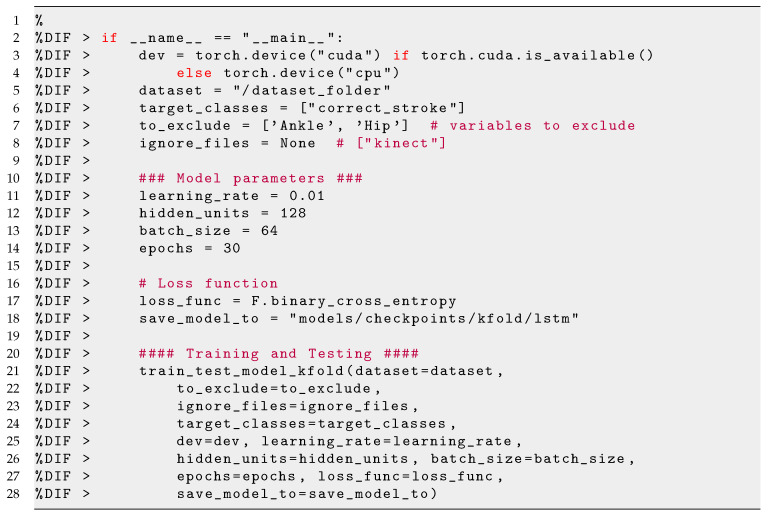
Code snippet for training the LSTM network.

### 3.8. Expert Interview

Since we managed to acquire only one table tennis coach, only one session was held. The whole session lasted around 40 min. During the session, we presented a set of questions with three blocks: (1) Demographic, (2) Training sessions, and (3) Design Feedback Requirements. Before proceeding to block (3), we described our study’s idea and the potential of a computer system that can help beginners improve their performance by providing feedback in real-time. We, therefore, provided an example by showing a short video of the Omron Forpheus (Omron Forpheus—https://www.omron.com/global/en/technology/information/forpheus/index.html, last accessed on 20 March 2021), an AI ping pong robot that is able to assist table tennis players during training by providing immediate feedback.

The first block explores the coach’s background in the table tennis domain. To be more specific, we asked about her experience of playing table tennis and coaching beginners. Next, we investigated the training drills prepared by the coach for beginners and how she assisted them during training sessions (e.g., rectifying common mistakes, type of feedback). The last block explores the coach’s conceptualisation towards a computer system that can assist the beginners during training sessions by providing feedback in real time.

## 4. Results

### 4.1. RQ1—Neural Networks’ Results

The dataset transformed into a tensor of shape (510, 51, 46) where 510 are the learning samples (260 short strokes + 250 long strokes), 46 the attributes from both smartphone and Kinect (6 and 40 attributes respectively), and 51 the fixed number of time updates (time-bins) for each attribute. Each stroke being annotated with three classes: smartphone sensors alone, Kinect sensors alone, and both sensors together. The distribution of classes is the short stroke (*correct_stroke*) with the binary classification of 0 and 1.

Every time a random split is done between training and validation, the results retrieved were different. The intuitive explanation is that the deficiency of the whole collected dataset, which is small in size. Therefore, we used cross-validation for the resampling procedure to evaluate the model on limited data. The procedure has a single parameter called “k” that refers to the number of groups that a given data sample is to be split into. As such, the procedure is called k-fold cross-validation. When a specific value for “k” is chosen, it may be used in place of “k” in reference to the model. In our case, it was 10-fold cross-validation. Consequently, k-fold cross-validation used limited data to estimate how the model is expected to perform in general when used to make predictions on data not used during the training of the model.

Table 1 shows the results of LSTM reporting for each target class with accuracy, precision, and recall. In the context of our study, accuracy is about how close the attempts are to the target, which is the short-stroke/correct stroke, and precision is about how close they are to each other. Recall identifies positive results to get a true positive rate.

In Table 1, we observed that the performance of the smartphone alone only achieved 51% of accuracy, which is poor as compared to the Kinect, which performs slightly better but still bad at 64% of accuracy, whereas the combination of both devices achieved similar results to the Kinect alone but with better precision at 73%. We observed that, by adding the number of attributes, the model improves.

### 4.2. RQ2—Expert Interview Results

The first block focuses on the background of the coach. Our only sample has 21 years of playing experience and roughly four years of coaching experience. The coach pointed out that, in general, beginners need more attention than advanced players during training sessions. Due to this, she prepared various training drills intended for the beginners, which leads to the second block.

Such drills include training the fundamental skills (i.e., forehand and backhand) repetitively before moving onto more advanced skills. Techniques observed, such as the execution of the strokes, the proper way of holding the racket and the movements. Doing these incorrectly considered common beginner mistakes that the coach has noticed in general. Hence, the coach preferred a personalised session with one beginner as she would be able to concentrate on the beginner’s performance and provide feedback instantly. In such a case, when the coach is not available during the training session, she suggested that the beginners should video record their performances to analyse them afterwards or watch and learn how the advanced players would perform the desired techniques.

For the last block, we explained the T3 and the potential of a computer system that can help beginners improve their performance by providing real-time feedback. Generally, the coach agreed but suggested that beginners should start with their table tennis class with human coaches for at least five sessions, which subsequently the computer system would complement their progress. Additionally, using the smartphone, the coach admitted that the system could be an alternative plan for beginners to practice when she is not available during the training session. As for the type of feedback, the coach preferred audio or sonification but not in real-time as it could be increasingly irritating for beginners when they make mistakes consecutively. Therefore, similar to our initial plan, she preferred an automated feedback system that fires if there are more than five mistakes detected in the last ten strokes, which would suit better. Furthermore, the coach thought that placing the smartphone in the pocket would affect the beginner’s performance.

Subsequently, we asked the coach what she would like the most and least about the system. For the former, she described that the system would improve and motivate the beginners’ performance when she, as the coach, is not around. As for the latter, since the fundamental techniques involve mostly on body motions, she mentioned that the system could be unfriendly and unnatural to disabled people, particularly those on wheelchairs. Lastly, when asked if she would include such a system in her training program, she agreed that this would help her teach more beginners concurrently and observe their performance better from far away.

## 5. Discussion

### 5.1. RQ1

RQ1 focused on exploring whether the smartphone sensors can classify forehand table tennis strokes as accurately as the Kinect. In Section 4.1, we observed that the LSTM performance (average of accuracy, precision, and recall) of the smartphone alone only achieved 51% classification accuracy, the Kinect with 64% accuracy, and the combination of both devices achieved a similar result to the Kinect but with better precision and recall at 73% and 61%, respectively. Based on these results, we reject RQ1. However, these results are the outcome of an LSTM approach. Lim et al. [25] also used the LSTM and achieved an accuracy of 93% but with a more complicated sensor setup built for a specific purpose. Consequently, we assume that our results are limited by only using generic sensors, such as i.e., smartphone sensors and the Kinect. Furthermore, earlier studies such as [26,27] have also used other approaches such as the SVM with varying results. Hence, it may be possible to get better results using different machine learning approaches with smartphone sensor data.

The precision (73%) and recall (61%) for the combined devices (Smartphone + Kinect) achieved the best results compared to the other two classes. A reasonable explanation would be the skeleton tracking from the Kinect’s depth camera sensor, as it tracks not only the participant’s joints but also the whole-body movements. However, the Kinect is not a commonly owned device and our intention is to develop a more accessible solution, which therefore resulted in us selecting the sensors in a smartphone which is owned by the majority of the population. We observed that combining the devices improves the performance of the smartphone. Thus, the smartphone can be complemented by other wearable devices such as a smartwatch to improve the performance.

Intuitively, the model is fed with data but also additional noise. This causes overfitting to happen [33], as the model learns the detail and noise in the training data to the extent that it negatively impacts the performance of the model on new data. Cross-validation was used as a preventative measure against overfitting (-Overfitting in Machine Learning: What It Is and How to Prevent It.—https://elitedatascience.com/overfitting-in-machine-learning, last accessed on 17 March 2021), but the results retrieved were still poor. Furthermore, it is not known whether 51% accuracy for the smartphone was achieved as a classification or by complete chance. Hence, in the context of our study, the smartphone sensors were unable to classify the strokes as accurately as the Kinect. Instinctively, this occurs due to the number of data samples collected were small in size and insufficient. Furthermore, these observations can’t be generalised due to only one sample/participant tested. In addition to this, positioning the smartphone device in the pocket could be one of the main factors that the performance was poor. Thus, adding more sensors, collecting more data samples, and positioning them differently (i.e., attached on the racket) could potentially improve the stroke recognition rate.

### 5.2. RQ2

RQ2 focuses on the coach’s conceptualisation of the system whether it could improve the beginners’ performance during training sessions. The interview results enabled us to better understand the coach’s perception of a real-time feedback system and how such technology can be used to train table tennis beginners in training, paving the path for this technology to provide feedback to table tennis beginners. Overall, the idea of the T3 system raised mixed feelings in the coach. On the one hand, the expert was sceptical about the extent of the effectiveness of the system in training. On the other hand, she acknowledged the difficulty of providing sufficient and effective feedback to the beginners and that the system could be beneficial to support the user. Since these technologies are relatively new in this domain, the scepticism of the expert is well placed. Furthermore, a precious contribution was retrieved from the coach, who mentioned that the system would work better with auditory feedback, which aligns with our system design. However, this modality would introduce an irritating experience to the beginners if given in real-time when they continuously make mistakes. After the explanation of our approach, which includes an automated system instead of real-time, the coach thinks that it could be useful not only for the beginners to train when she’s not around but also for herself to include it in her training program. Some additional suggestions were given by the coach to improve the system further. Recommendations include the use of audio feedback instead of visual or haptic mode, so as to not overload the beginner. These suggestions reinforce and enrich the idea of implementing a feedback mechanism that can efficiently assist table tennis beginners during training.

## 6. Conclusions

In this paper, we explored an approach for classifying the forehand table tennis strokes using multimodal data and neural networks. We designed the T3, a multi-sensor setup consisting of a smartphone device with built-in motion sensors and the Kinect depth camera sensor. We validated the collected multimodal data three times, observing the performances of (1) only smartphone sensors, (2) only the Kinect, and (3) both devices combined. We observed, within our context, that smartphone sensors by themselves are unable to perform better than the Kinect. In addition, it is likely that the smartphone sensors are able to classify the strokes by complete chance due to 51% of accuracy. However, the performance improves when both of the devices are combined. Of course, our study is by no means comprehensive and many limitations exist (see Section 7), which prevents us from drawing a conclusion for our results. Therefore, in this limited study, we conclude, as many other studies have, that more sensors are needed, in addition to the smartphone sensors for classifying forehand stroke in table tennis more accurately.

Our intention with the fore-mentioned technology was to use it to provide feedback to the employees. To facilitate this further, we organised an expert interview with a table tennis coach to explore the coach’s perception of a computer-based training system that provides feedback when mistakes are detected during training. The coach expressed her doubt about these technologies, especially AI, since they are relatively new and not widely adopted. However, the coach agrees that the system could indeed help beginners train their fundamental techniques and act as a support to further improve the learning outcome.

## 7. Limitations

Several shortcomings were encountered during the study from multiple perspectives. Primarily, the global pandemic of COVID-19 has restricted our objectives from collecting more data samples from the participant. Therefore, the LSTM results retrieved were poor due to the insufficiency of data (only 260 short strokes and 250 long strokes). Moreover, the Kinect only recorded the upper half of the participant’s body. Full-body tracking could have resulted in higher accuracy for the Kinect. The data from the various placements of the smartphone were manually visually judged, which could have led to unknown inaccuracies.

In addition, since strokes’ execution relies mainly on the body motion, other players with different body structures or playing styles would introduce additional uncertainty, which may affect the system’s accuracy. Furthermore, the COVID-19 limitations prevented us from conducting a user study, which was initially planned for beginners with the inclusion of a real-time feedback system. We, therefore, changed our approach into organising online expert interviews for the collection of qualitative data as an alternative. However, despite doing so, we managed to acquire only one coach after multiple search attempts.

Ideally, in our study, placing the smartphone device in the pocket helps the participant perform the strokes conveniently. However, having it inside for a long continuous session could turn out to be increasingly inconvenient due to its bulkiness. In addition, players typically wear loose shorts during the game, which removes the possibility to use the smartphone in actual club based training. A long session was also not possible due to the limitations of the LearningHub, which is only designed for short activities, and the mental and physical effort required to constantly perform short strokes repeatedly. A further limitation is the use of only one smartphone device. Newer smartphones provide more built-in sensors, and in some cases, the sensor layouts could differ from the smartphone used for this study.

## 8. Future Work

Despite not achieving the expected results, we think that T3 still has unexplored potential. Thus, several ideas and improvements should be further explored before this system can be used in training. The combination of the smartphone and Kinect shows the highest accuracy and precision, leading to more accurate and effective feedback. Thus, including wearable technology such as a smartwatch will potentially improve the accuracy of the system. In addition, using a smartwatch mitigates a bulky smartphone’s limitations while performing the strokes, which enables the expert or beginner to execute the stroke naturally.

This approach can also be applied in a more distinctive scenario for disabled table tennis players on wheelchairs, as suggested by the table tennis coach in our expert interview. Their strokes’ execution could differ from the regular players due to the limited movements, especially from their lower body parts. This means that the coach needs to put more time and effort into training them, as they require personal attention. Hence, our approach will likely be beneficial for helping disabled players improve their table tennis skills.

Further work could include further exploitation of data. With sufficient data samples, it is possible to improve the accuracy of the a single device or gain higher accuracy with multiple sensors. This will prove beneficial for implementing an automated (auditory) feedback system to intervene the beginners when incorrect techniques are detected. A follow-up user study should also be conducted to investigate the effectiveness of provided feedback interventions for beginners during table tennis training. Furthermore, since our study focuses on classifying the most fundamental table tennis technique, the identification of the other three strokes (forehand push, backhand drive, and backhand push) could also be further explored. This gives the flexibility to the beginners to improve their overall fundamental techniques at their own pace.

## Figures and Tables

**Figure 1 sensors-21-03121-f001:**
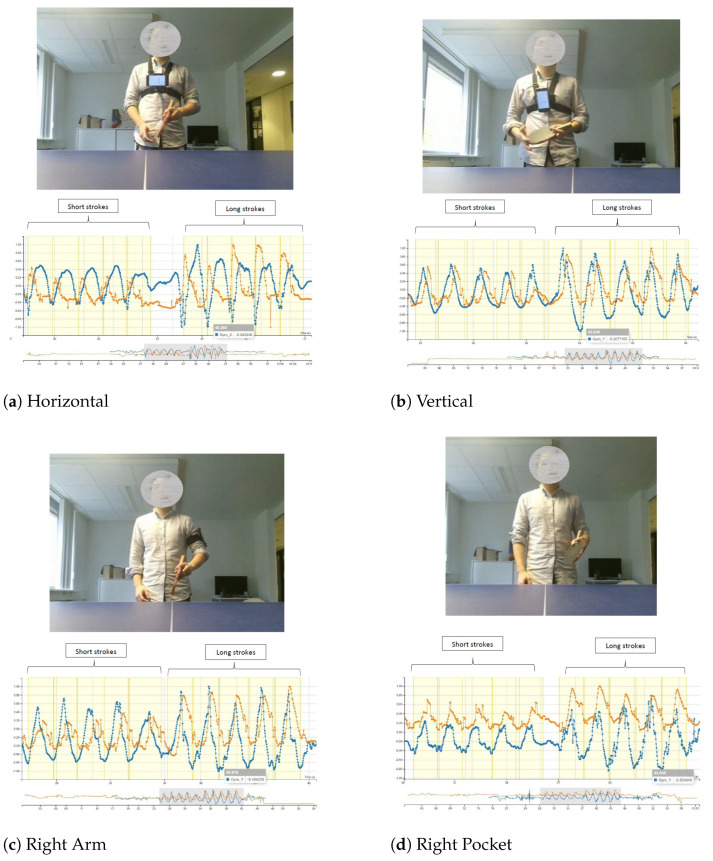
Results of the stroke performance from four different positions with the following attributes: (Blue Lines) Gyro_Y; (Orange Lines) HandRight_Y.

**Figure 2 sensors-21-03121-f002:**
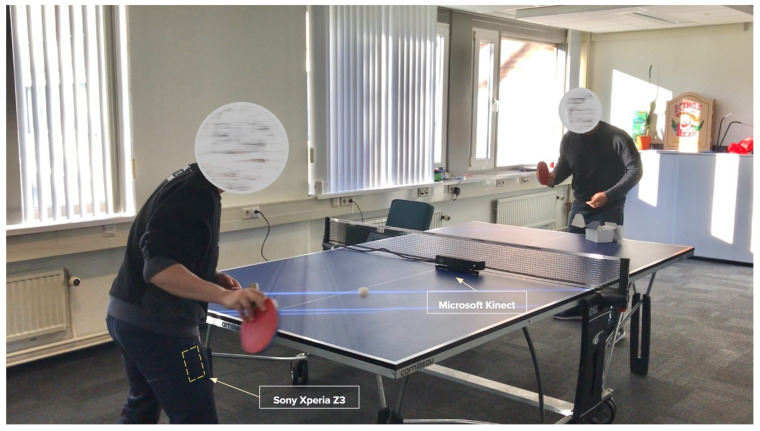
Data collection setup.

**Figure 3 sensors-21-03121-f003:**
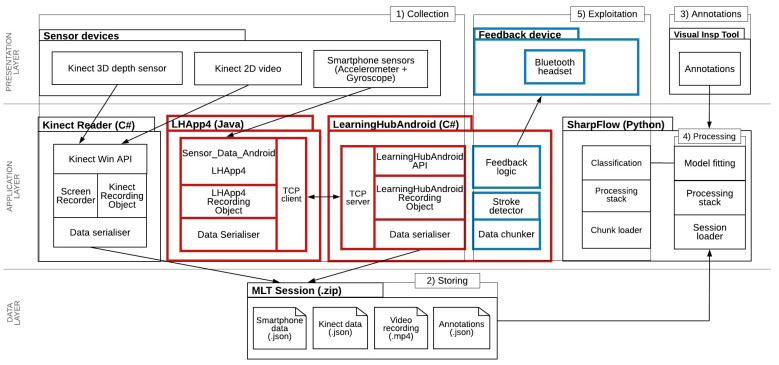
The Project Architecture with the following elements: (Red lines/boxes) New components developed for this study. (Blue lines/boxes) Future work.

**Figure 4 sensors-21-03121-f004:**
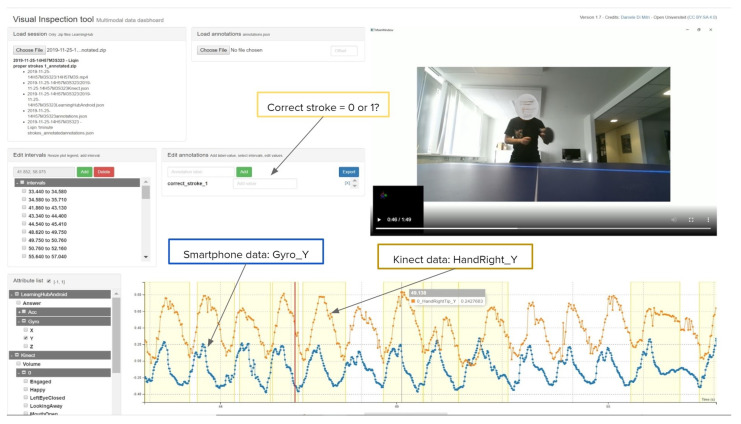
Screenshot of the VIT.

**Figure 5 sensors-21-03121-f005:**
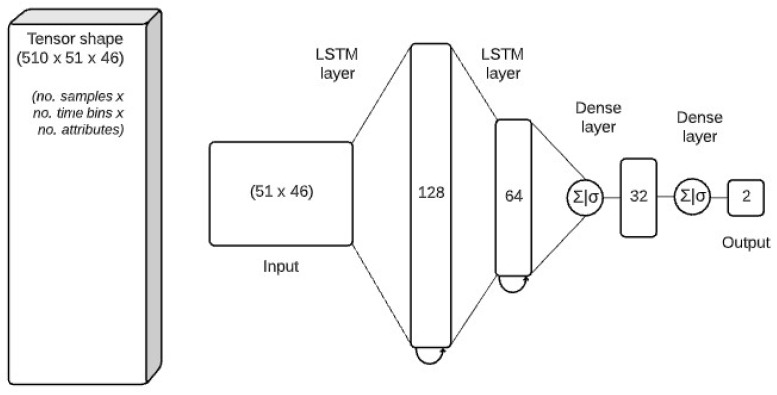
The Neural Network architecture for the stroke detection.

**Table 1 sensors-21-03121-t001:** The LSTM results in terms of classification Accuracy, Precision, and Recall.

Class	Accuracy	Precision	Recall
Smartphone	0.51039	0.39364	0.48636
Kinect	**0.63896**	0.54607	0.55135
Smartphone + Kinect	0.62078	**0.72933**	**0.61111**

## Data Availability

The repositories of the components used for this study can be found in this article with links provided. We are unable to release the dataset due to proper anonymisation has not yet been adopted, as the participant is clearly identifiable from the videos recorded by the depth camera sensor. However, we are considering to do so in the near future.

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
