# Peer review of "Table Tennis Tutor: Forehand Strokes Classification Based on Multimodal Data and Neural Networks"

_sensors, 2021, doi:10.3390/s21093121_

Round 1

Reviewer 1 Report

Authors introduced the Table Tennis Tutor (T3), a multi-sensor system consisting of a smartphone device with its built-in sensors for collecting motion data and a Microsoft Kinect for tracking body position of a player. Authors successfully classified correct and incorrect forehand stroke by player using a recurrent neural network. 

It is well written and organized paper and can be improve. My comments are as follow:

  1. To make your data more general and realistic, it would be great if dataset is collected from multiple users for both RQ1 and RQ2.
  2. Please provide Neural Network architecture diagram for better understanding. 
  3. During the Neural Network training, what was stopping criteria. (Why stopped at 30 epochs?)
  4. In Table 1: Can you please bold the best results and discuss: Why they are best?

Minor:

  1. Figure 10 is missing.
  2. Few grammar and typing mistakes need to be rectify. 

Recommendation:

  1. It would be great if authors provide dataset publicly available for future works.

Reviewer 2 Report

The paper presents a tool for classification of table tennis strokes, to be used in training sessions of beginners players, when the coach is not available.

The underlaying idea is interesting and could be useful and re-applied in multiple fields, especially during times of isolation as the one caused by the COVID-19 pandemic.

However there are major issues in the presentation of the paper, as an example:

  • Only one subject is not a valid dataset and is not representative.
  • Moreover, the ground truth for the "wrong strokes" are simulated, it would have been better to use a set of actual beginners recorded during trainings as simulated activities are proved (although in other classification probes) to be non-representative of true actions.
  • The related work is not sufficient. The authors reference more than one work that could be directly compared with the proposal, however, it is not clear in which way the proposed tool is different from those proposals (especially [7] by the description on the paper).
  • There is no form of comparison with the literature, a comparison with any of the other mentioned solutions would be greatly beneficial for the soundness of the proposed approach.

There are other minor issues, such as missing references and typing errors noted in the attached version of the paper in forms of notes and highlights.

Finally, the images and listings have some layout issues and captions missing, but this might be a problem related to the submission platform

Reviewer 3 Report

The document provides preliminary research on detecting  forehand motion during table tennis game. The authors, from data set recording an experienced and nonexperienced table tennis players who performing 260 short forehand strokes (correct) and mimicking 250 long forehand strokes (mistake). They analyzed and notated multimodal data to train a neural network that classifies correct and incorrect strokes. To test the accuracy level depending on various smart sensors. The study confirmed three combinations: smartphone sensors only, Kinect sensor and both devices combined .  The article is written in good English (however, some sentences may be shorter), the structure of the work is correct, and the subject falls within the scope of Sensors journal.

  • The title of the article is "Table Tennis Tutor: Forehand Strokes Classification based on Multimodal Data and Neural Networks." and even the method of multimodal signal analysis is briefly described.
  • The algorithm is based  on the  method from source the  publication  Di  Mitri, D.; Schneider, J.; Specht, M.; Drachsler, H. “The Big Five: Addressing Recurrent Multimodal Learning Data Challenges. Proceeding Companions of the 8th International Conference on Learning Analytics and Knowledge”, checks whether smartphone sensors can classify table tennis movments  as accurately, dataset from Kinect and Kinect to supported by a gyroscope  dataset from smartphone.
  • Using Kinect data as a benchmark for performance comparison. In my opinion, the conclusions should be rewrite, because it is not known whether the Accuracy of 0.51039 for the smart phone class was actually achieved as a result of classification or accidentally.
  • Referring to the section "Data collection procedure" which does not raise doubts about data collecting originality (bravo for photos to facilitate the image of the measuring system).
  • There is no discussion of code changes in relation to the original coding of the downloaded publication Di Mitri at al in the introduction. The authors rightly noted  that "This study evaluates the stroke detection system of the T3 to train table tennis using smartphone sensors. To achieve this, a workflow consisting of steps for collecting and processing the data recommended by Di Mitri at al  is followed. "

I could not find a description of these differing and evaluated code in the text. Was the source code from the publication of Di Mitri  at al, unchanged?  

  • Activities such as playing tennis are highly complex skills that require a lot of training. In my opinion - omit in the article the assisted or supporting aspect. Focus on describing how  the data was classified?

Needs to be improved:

1.Accurately describe the code changes made in relation to the source coding. Suggest an algorithm  scheme or  print  screen of your own code.

2.No discussion, drawing or other visual representation, how neural network was learned or trained .

3.No discussion and defense of low score for Accurancy  0.51039. There is no  visual analysis or statistical accuracy of the results obtained.

4. The results presented in the Table 1. "The LSTM results in terms of classification Accuracy, Precision, and Recall. " - should be expanded.

In my opinion presented result are, not sufficient.  But one, can do be defended with appropriate arguments.

5. The document seeks to describe the issues of supporting the learning of tennis and the classification of multimodal data in a balanced way. Which, in my opinion, somewhat blurres the whole main idea of the study .

Positive:

A unique and completely original data set.

Interesting use of knowledge.

Good English.

I hope I helped. Good luck with your research!

Round 2

Reviewer 1 Report

Authors updated the paper as per recommendations.

I accept the paper in present form.

Reviewer 2 Report

I appreciate the effort of the authors in revising the paper following my comments.

It would be interesting to see an extension of the paper featuring a richer dataset with more subjects. Would it be possible to move the Kinect to the side and therefore register actual games?